# Population and Housing (Mis)match in Lisbon, 1981–2018. A Challenge for an Aging Society

Nachatter Singh Garha *[iD] and Alda Botelho Azevedo [iD]

Instituto de Ciências Sociais, Universidade de Lisboa, 1649-004 Lisboa, Portugal; alda.azevedo@ics.ulisboa.pt
* Correspondence: nachatter.singh@ics.ulisboa.pt

**Abstract:** Over the past four decades, Lisbon's population has witnessed some notable changes in its growth, age structure, conjugal patterns and ethnic diversity. These changes have affected the size, composition and structure of households, which in turn modified the demand for housing in Lisbon. At the same time, some changes were also made to the housing stock, especially in size, but also in the replacement of small apartment buildings with large residential apartment blocks. However, the discrepancies between demographic changes and the housing stock have created new challenges for the housing market in Lisbon to provide adequate housing to all households in the city. Using data from the 1981–2011 census obtained from IPUMS International, population estimates from the National Institute of Statistics (INE) and other secondary sources, this article identifies and measures the magnitude of (mis)matches between existing housing stock and housing needs of the population in Lisbon. Theoretically, this article adds to the knowledge about the relationship between population and housing dynamics in large cities. Empirically, it provides evidence on the existence and magnitude of (mis)matches in Lisbon's housing market and explores the underlying dynamics behind them. Lastly, it offers some policy tools and recommendations to address these (mis)matches.

**Keywords:** demographics; family dynamics; housing demand; housing stock; Lisbon

## 1. Introduction

In recent decades, adequate housing for all has become a major concern in many European cities, which are experiencing demographic changes induced by the accelerated aging process and massive immigration from different countries outside the European Union (Pittini et al. 2017). Most of these housing problems are directly or indirectly related to the changes in the demographics and speculative construction during the last few decades without taking into consideration the future needs of the population. Whilst there are many studies on the interaction between financial and property bubbles, the relationship between housing and demographic cycles is still understudied (Myers and Ryu 2008; Myers and Pitkin 2009). In this article, we are going to focus on this interrelation in Lisbon.

Like other big cities in Southern Europe, such as Madrid, Barcelona, Milan and Athens, in recent years, Lisbon, the capital city and the largest urban agglomeration in Portugal, has suffered several issues related to access to adequate housing (Arbaci 2019; Branco and Sónia 2018; Tulumello and Allegretti 2020). Since 1981, the municipality of Lisbon has witnessed a significant decline and rapid aging of its population. In 1981, the total population of Lisbon was 807 thousand people, about 9% of the total population of Portugal, and the average age of the population was 37.7 years. In 2019, according to estimates by the National Statistics Institute (INE), its population size decreased to 509.5 thousand residents (4.9% of the total population) and the average age increased to 45.9 years. During this time, the typical household size and structure also changed with the near disappearance of complex households and the increasing number of one-person households. Although housing has been, since the second half of the twentieth century, one of the main concerns for the local

administration in Lisbon, the massive construction projects started in the late 1970s (Agarez 2018) disregarded important demographic trends in the city, namely out-migration (both national and international) and decreasing fertility levels. As a result, Lisbon became a good example to demonstrate how in big cities, demographic dynamics and the housing stock can evolve in opposite directions, leading to an excess of housing stock and yet not meeting the housing needs of the population.

Previous studies have shown a two-sided relationship between population dynamics and the housing market in big cities (Mulder 2006). Some researchers have argued that demographic events such as births, migration, union formation and dissolution, disability and death are crucial factors in shaping the supply and demand for housing (Clark and Dieleman 1996; Myers 1990a; Azevedo 2016), while others have highlighted the impact of the size and quality of housing stock and the prevailing tenure regime on the demographic structure of different cities (Castiglioni and Dalla Zuanna 1994; Pinnelli 1995). A sufficient supply of adequate and affordable housing (both for sale and for rent) helps nest-leaving young adults to form new family households and, later, to fulfil their fertility intentions (Vignoli et al. 2013). It also prevents young couples from leaving their hometown and attracts immigrants who increase the size of the population and often make it younger and ethnically diverse (Fransson 2000).

The main objective of this study is to identify and measure the magnitude of (mis)matches between the existing housing stock and the housing needs of the population of Lisbon. Our testable hypothesis is that the differentiated demographic and housing trends over the last four decades have created several (mis)matches that affect different age and socio-economic groups in Lisbon differently. Thus, conceptually, this work follows the framework used earlier by Arestis and Gonzalez-Martinez (2017). Methodologically, using a representative sample of Lisbon municipality population and housing as a primary data source (microdata from the IPUMS-International database), we first observed the changes in the structure and household composition of Lisbon's population in the period from 1981 to 2011. Next, we focused on the changes in the size and characteristics of the housing stock and the prevailing tenancy regime in the city. Finally, based on the cross-reference of the demographic and housing stock characteristics, we have identified several (mis)matches in the housing market of Lisbon.

This article makes a triple contribution. It adds to the theoretical knowledge on the relationship between population and housing dynamics in big European cities; it provides empirical evidence on the existence and magnitude of (mis)matches in Lisbon's housing market; and lastly, it provides some policy tools and recommendations to address these (mis)matches. Given that Lisbon shares many of its sociodemographic characteristics (low fertility, aging, erosion of complex households and immigration) and housing issues (size and age of housing stock) with other big European cities, our recommendations can be extrapolated to frame new housing policies in other big cities.

The article is structured as follows. Section 2 presents a detailed review of existing studies exploring the relationship between demographics and housing. Section 3 describes the data sources and methods used in this study. Section 4 analyses population and household structure, size and quality of Lisbon's housing stock. It also highlights the mismatches between population and housing dynamics in Lisbon. Finally, Section 5 presents some concluding remarks and a discussion regarding the main aspects of these mismatches and suggests some measures to deal with them.

## 2. Literature Review

*The Two-Sided Relationship between Population Dynamics and Housing Market*

In the late 1960s, Campbell (1963, 1966) analyzed the relationship between demographic swings and housing construction cycles and showed the impact of irregularities in the demographic structure on aggregate demand for housing in the United States. Later, in his classic work on *Housing Demography*, Myers demonstrated that "at different geographic scales, demographic changes provide a solid explanation for many residential processes

that must be taken as a key aspect in the analysis of housing systems" (Myers 1990b, p. 306). More recently, Mulder (2006) explained the links between housing and population as a two-sided relationship, in which the population influences housing through the demand created by nest-leaving young adults, marriage or consensual union, childbirth, separation or divorce, and housing influences the geographic distribution of people and households through the attraction or deterrence of migrants, facilitating or hindering nest-leaving of young adults and keeping in place or pushing away the resident population (Figure 1).

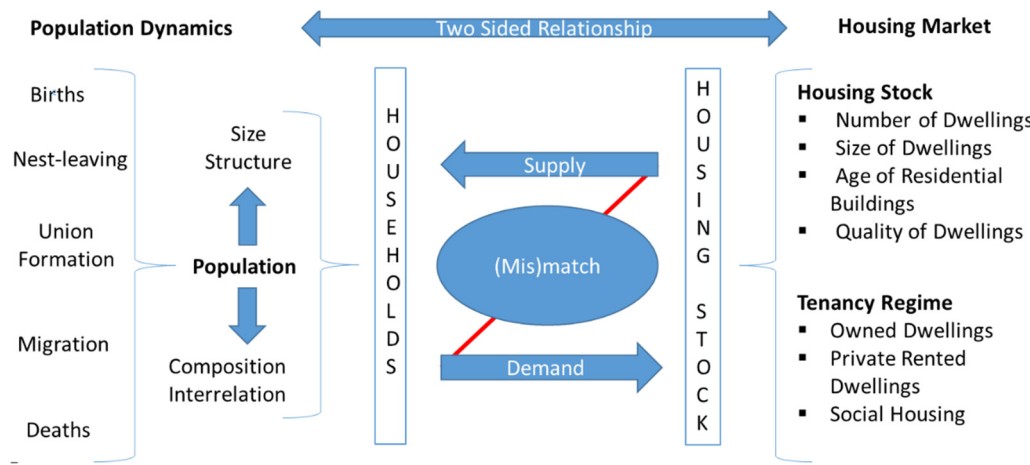

**Figure 1.** Two-sided relationship between population dynamics and housing market. Source: Own elaboration.

Among demographic factors, age structure is considered the most important factor that determines the interaction between population and housing behavior (Myers 1990b; Myers and Pitkin 2009). The demand for housing changes with age, both in qualitative and quantitative terms (Marcos and Módenes 2019). This change, at an aggregate level, determines the total demand for housing units in the future. Campbell (1963, 1966) in his analysis of population dynamics and construction cycles in the United States, has emphasized the importance of age structure in shaping the demand and supply of houses and its impact on house headship and ownership rates. In the young population, the flow of new households is high, which in turn increases the demand for new housing units. In contrast, in an aging population, the balance between the formation of new households by young adults and the dissolution of households with the death of the elderly reaches equilibrium, which in turn offsets the demand for new housing units (Myers et al. 2002). In their study of OECD countries, Lindh and Malmberg (1999, 2008) demonstrated that there is a close relationship between the aging process and the reduction in demand for new housing. In an aging population, such as in Lisbon (Bandeira et al. 2014), as more people die or move to retirement homes, more apartments become vacant. This can lead to a situation in which the supply of housing units exceeds demand, resulting into underutilization of housing stock and a fall in home equity, in the sense that new buyers are willing to pay less due to the oversupply in the housing market (Pitkin and Myers 2008). Disparities in the size of different generations as they progress over time also affect the demand and supply of housing (Myers 1999; Myers and Ryu 2008). As cohorts with a large population enter adulthood, the demand for housing increases, and as they age and begin to die or move into collective homes, they leave a large number of vacant apartments that increases the supply of houses in the housing market.

Some researchers have argued that the number, size and composition of households are more important factors then the total size of population in determining future housing needs (Mulder 2006). Households are very dynamic units. As people grow and form relationships, they often move from one household type to another, which in turn changes their housing needs. The high cost of construction and the long life cycle of dwellings (once

built dwellings can be used for several decades) make the housing stock less flexible to meet the requirements of highly dynamic population and households. People often find it difficult to move from one home to another with every change in their life course, such as marriage, childbirth, disability and the death of the partner, due to financial constraints and also attachment to their existing homes and neighborhoods. This urge of "aging in place" (Frey 2007) reduces household's mobility from one dwelling to another, making the housing market less responsive to changes in demographic and household structure, and creates mismatches.

Nest-leaving of young adults is one of the most important steps in the formation of new households (Fransson 2000). Young adults leave their parent's home for a variety of reasons, including education, work, independence or union formation, creating demand for new affordable housing (Azevedo 2016). Those who leave their parent's home for education and work have few choices and are more likely to accept substandard houses with relatively high rents; however, those who migrate for independence or union formation (marriage or consensual union) have more options and they can wait until they get a suitable and affordable house (Mulder 2006). Their choice of housing tenures is subject to several constraints such as housing availability and affordability, income and cultural aspects related to young people's preferences and expectations (Lennartz et al. 2016; Fuster et al. 2019). At the same time, the supply of affordable and easily accessible houses also affects the nest-leaving age of young adults (Mandic 2008). In their recent study of millennials in the United States, Bleemer et al. (2014) showed that young adults are more likely to delay nest-leaving because of the high price of housing, unemployment and student loans. Similarly, Ermisch and Di Salvo (1997) explained that high rents or mortgages, a limited supply of affordable houses and high rates of homeownership affect nest-leaving prospects and make it particularly difficult for young adults to form their own households. After nest-leaving, marriages or union formations also contribute to increasing the demand for houses. When people plan to get married, they often like to secure a suitable place for themselves first. The number of marriages in a year has a positive effect on the demand for new houses. Similarly, the supply of adequate and affordable houses also affects the occurrence of marriages. People tend to postpone marriages or union formations when they do not find adequate housing (Clark 2012).

The birth of a child in a household reduces the per capita space available to all members in a dwelling. The lack of living space often compels households to search for larger dwellings, increasing their demand in the housing market (Azevedo et al. 2019a, 2019b). In their study, Feijten and Mulder (2002) showed that couples in the Netherlands frequently move into a single-family home or an owner-occupied home, after or shortly before the birth of a child. The number of children born in a given year has both short- and long-term effects on the housing market. In the short-term, the high number of annual births increases the aggregate demand for family dwellings, which in turn increases their purchase price and monthly rents. In the long-term, the demand for small houses increases as this large cohort reaches the nest-leaving age at the same time, around 20 years after birth. The supply of suitable and affordable houses also affects the decision to have the first and subsequent child. In countries where access to suitable housing is difficult, couples often delay childbearing, which in turn reduces the desired number of children and the total number of births (Castiglioni and Dalla Zuanna 1994; Pinnelli 1995). Some studies have also shown that housing plays an important part in relation to divorce. Homeowners in Germany (Wagner 1997), Australia (Bracher et al. 1993) and Finland (Jalovaara 2002) were also found to be less likely to divorce than renters. Death in a household also affects the housing market in several ways. In a one-person household, death adds another dwelling to the housing stock. Death of a household member, such as a partner, may also indirectly affect the availability of dwelling, as the remaining household member may face financial constraints to maintain the dwelling (Fransson 2000).

In most European countries, immigration has become an important determinant behind all demographic changes (Zaiceva and Zimmermann 2016). Immigration creates

demand for dwellings and emigration contributes to their supply in the housing market. In his seminal research, Campbell (1966, p. 117) highlighted the importance of immigration in increasing the demand for rental houses in the United States. As migration affects the demand for housing, the availability and scarcity of houses also affects migration. The availability of adequate houses for rent may attract migrants or prevent emigration, while the scarcity of houses may prevent migrants from entering and encourages the emigration of young adults who wants to leave their parent's home (Mulder 2006). Similarly, the proportion of immigrant population also affects the home-ownership rate in big cities. There is a huge gap between immigrants and natives with respect to homeownership (Borjas 2002). In their research, Myers and Liu (2005) showed that upon arrival, immigrants have lower homeownership rates on average than people born in the United States; however, as length of stay increases, homeownership among immigrants also increases. According to Helderman et al. (2006), homeowners are less likely to migrate than renters, as the transaction costs of moving are much higher for owners than for renters. This means that the higher level of homeownership could seriously hamper the spatial mobility of people from one dwelling to another or from one place to another (McCarthy et al. 2001).

In addition to the demographic factors mentioned above, other socioeconomic and political factors such as household income, labor market conditions, ethnicity, availability of housing credit, financial institutions, economic cycles and policy conditions also have important influence on the housing market (Herbert et al. 2005; Arestis and Gonzalez-Martinez 2019). However, this article focuses on the two-sided relationship between demographic factors and housing characteristics that affect the functioning of the housing market in Lisbon.

## 3. Data Sources and Method

For this article, the primary data on the demographic and housing characteristics of the Lisbon population have been extracted from the IPUMS-International (IPUMS 2020) database (the last four rounds of the census, i.e., 1981, 1991, 2001 and 2011). From the census records, information on demographic factors (size, age-structure, sex-composition, civil status, household structure and composition, and migration status of population) and the characteristics of dwellings (number and size of dwellings, age of residential buildings, utilities available in different dwellings, and tenancy regime) is obtained to explore the changes in the demography and the housing market in the municipality of Lisbon. All information is available and representative of the resident population at the municipal level. In addition to the census data, information on the recent demographic trends (2011–2018) has been collected from municipal records published online by INE Portugal.[1]

A detailed exploratory data analysis (Tukey 1977) of the changes in the population structure and household composition of Lisbon over the last four decades (1981–2019) has been carried out and then compared with changes in the size and characteristics of the housing stock and tenure regime in the city. This helped us to identify and measure the magnitude of the main (mis)matches in the Lisbon housing market.

## 4. Results and Discussion

### 4.1. Population, Households and Housing in Lisbon, 1981–2011

Population and household dynamics are the main aspects of urban change worldwide (Alonso 1980). In order to study the relationship between population dynamics and housing market in Lisbon, it is essential to explore the changes in the structure and composition of population, the changes in the number, size and typology of households, and the evolution of the housing stock and tenancy regime in the last four decades.

---

[1] More information and access to data of IPUMS-International database can be found at IPUMS International and on municipal records of INE Portugal at Portal do INE.

4.1.1. Population Dynamics

For a long time, Lisbon has been the most densely populated city in Portugal (Veiga 2004). During the twentieth century, the size, structure and composition of its population have witnessed notable changes towards depopulation, aging and ethnic diversification. In 1961, its estimated population was 802.2 thousand, which represented 9.7% of the total population in Portugal. In 1981, the total population reached to its maximum, i.e., 807.9 thousand and then began to decrease drastically, i.e., −17.9% in the 1980s and −15.3% in the 1990s. It reached to 562.1 thousand in 2001. According to the INE, in 2019, the estimated population of Lisbon was 509.5 thousand and its share in the total population was 4.95%, which is the lowest figure in the last century (Table 1). In this article, we have focused on the demographic changes that have occurred since 1981, because it represents the maximum number of resident population in Lisbon.

**Table 1.** Population size, growth rate and proportion of the total Portuguese population in Lisbon, 1900–2019.

| Year | Lisbon Population | | | | Live Births (in ′000) | | Life Expectancy at Birth (e0) | Mean Age at Childbearing | Total Fertility Rate (TFR) |
|---|---|---|---|---|---|---|---|---|---|
| | Population (in ′000) | % of Total Population | Decadal Growth (%) | Immigrants (%) | Lisbon | Portugal | Portugal | Portugal | Portugal |
| 1961 | 802.2 | 9.7 | −5.2 | | 20.4 | 217.5 | 63.9 | 29.6 | 3.2 |
| 1971 | 760.2 | 9.4 | 6.3 | | 15.7 | 181.2 | 66.9 | 29.0 | 3.0 |
| 1981 | 807.9 | 8.7 | −17.9 | 6.7 | 10.8 | 152.1 | 71.7 | 27.2 | 2.1 |
| 1991 | 663.4 | 7.1 | −15.3 | 6.5 | 6.3 | 116.3 | 74.1 | 27.5 | 1.6 |
| 2001 | 562.1 | 5.4 | −3.5 | 8.5 | 5.8 | 112.8 | 76.7 | 28.7 | 1.5 |
| 2011 | 542.4 | 5.1 | −6.1 | 12.1 | 4.8 | 96.9 | 79.7 | 30.1 | 1.4 |
| 2019 | 509.5 | 5.0 | | | 4.3 | 86.2 | 80.6 | 31.2 | 1.4 |

Source: Own elaboration, with data from census records 1900–2011 and population estimates 2018, Lisbon (INE).

Since 1981, the accelerated depopulation has become a main feature of Lisbon's demography. In the period between 1981 and 2019, Lisbon has lost 298.4 thousand people, approximately 37.2% of its residents. A comparison of the population structure of Lisbon in 1981 and 2018 shows that most of the depopulation occurred in the age-groups of 0–70 years (−328.4 thousand). However, positive growth is recorded in the age group of 70+ years (+30 thousand). This shows the continuous aging of Lisbon's population.

According to the census microdata, the mean age of the population in 1981 was 37.7 years, much higher than the national average, i.e., 31.1 years. It increased to 41.1 years in 1991 and 44.3 years in 2001 due to the sustained decline in fertility levels and increased life expectancy. Between 2001 and 2011, this trend slowed down with sustained national and international immigration flows to the city. In 2011, it reached 44.5 years (42 for men and 46.5 for women), which was still higher than the national average (41.8 years), but the difference narrowed from 6.6 years in 1981 to 2.7 years in 2011. More recently, in 2019, it was 45.9 years (42.8 years for men and 48.6 years for women), which demonstrates the continuity of the aging process. Contrary to what happened at the national level in the 1960s, where the emigration of young people was the main driver of population aging (Bandeira et al. 2014), in Lisbon, aging was mainly due to the increase in the life expectancy at birth and the rapid decline in fertility. Improvements in living conditions, medical care and health services contributed to reducing mortality and extending life expectancy. Life expectancy at birth (e0) has increased remarkably from 63.3 years in 1960 to 80.9 years in 2018 (Pordata 2020). On average, women in Portugal live 6.5 years more than men. This gender difference has been more or less constant for the past half century. Regarding fertility, the total fertility rate (TFR), i.e., the average number of children per women, in Portugal, drastically decreased from 3.2 children per women in 1960 to 1.4 children in 2018 (Ibid). The fertility decline in the Lisbon metropolitan area was lower than the rest of Portugal (1.7 children per women in 2018), which was partly due to the concentration of

the immigrant population in Lisbon that has maintained a higher TFR compared to the native Portuguese population (Oliveira and Gomes 2014; Sousa-Gomes et al. 2016).

The speed of the aging process is also visible from the changing proportion of different age groups in the total population. The proportion of children (0–14), young adults (15–29) and adults (30–64) in the total population has decreased from 18.8%, 21.8% and 45.2% in 1981 to 16.3%, 12.1% and 43.4% in 2019, respectively. However, the proportion of the elderly (65 years and over) has increased from 14.1% in 1981 to 28.3% in 2019.

In addition to structural changes, since 1981, the population of Lisbon has also become increasingly diverse with the sustained influx of people from different EU and non-EU countries (Malheiros and Fonseca 2011). The total number of foreign-born residents in Lisbon has increased from 49.7 thousand in 1981 to 53.5 thousand in 2011. Despite a small increase in the actual number, the share of foreign-born residents has increased from 6.7% in 1981 to 12.1% in 2011. Most of the immigrant population is concentrated in the 25–50 age-groups. It has reduced the depopulation rate in Lisbon in these age groups. The immigrant population has also diversified significantly with the arrival of immigrants from countries that have no colonial or cultural ties to Portugal. In 1981, most of the immigrant population in Lisbon originated from former Portuguese colonies (Angola, Mozambique, Cape Verde, Brazil and India) and neighboring countries (Spain and France). In 2011, immigrants from some new countries of origin, like China, Romania, Guinea-Bissau and Nepal, constituted a considerable part of the total immigrant population in Portugal.

In short, we can conclude that the past four decades have shown that Lisbon is going through a process of accelerated depopulation, aging and ethnic diversification. Since these changes in demographic structure and composition have a significant impact on the formation and dissolution of households, which in turn affects the demand and supply of houses, it is important to analyze the impact of abovementioned demographic changes on the transformation of the household structure and the housing needs of the Lisbon population.

### 4.1.2. Household Dynamics

The demand for housing is not only determined by the number of people, but also by the number of households in a city (Mulder 2006). Changes in the formation, size, structure and composition of households play an important role in shaping future supply and demand for housing. There are four main forms of household formation: nest-leaving of young adults, marriage/union formation, divorce or separation, and older people moving in with their children or nursing homes. The distribution of population in different household types, such as one-person, couple with or without children, single parent and complex, is affected by numerous factors, including the age structure of the population, the level of union formation and dissolution, the average number of children per family, the cultural norms about housing patterns and the housing system.

The aforementioned demographic changes have a notable impact on the number, size and structure of households in Lisbon. According to census data, in 1981 the total number of households in Lisbon was 286 thousand and the average size was 2.8 people per household. It decreased to 234.5 thousand in 2001, with an average size of 2.4 people per household. After 2001, despite the decrease in the total population, the total number of households increased, reaching 243.9 thousand in 2011. Although the average size of households decreased to 2.2 person. A comparison between the decrease in the number of households and the population size in Lisbon between 1981 and 2011 shows that the decrease in population (32.1%) was much greater than the decrease in the number of households (14.7%). Since 1990, the number of households has remained stagnant, yet the population is steadily declining. The main driving forces underlying the decline in average household size are: the increase in the number of *one-person* households and constant low fertility over many decades.

In addition to the decline in the number and average size of households, the structure of households has also undergone notable changes in recent decades. In 1981, 30.5% of

the total population of Lisbon lived in *complex* households, where they shared dwellings with members of their extended family or others. In 2011, it decreased to 18.2%. However, the proportion of *one-person* households increased from 8.1% in 1981 to 14.7% in 2011. This increase in the number of *one-person* households is mainly due to the increase in life expectancy and the tendency among young adults to spend some time alone in small apartments, before entering into relationships with others. The proportion of *single-parent* households also increased from 4.97% in 1981 to 9.91% in 2011. Most importantly, the share of *couples without children* has increased from 12.8% in 1981 to 19.9% in 2011. During the same period, the proportion of *couples with children*, which is still the most numerous household type in Lisbon, decreased from 42.04% in 1981 to 35.7% in 2011. All of these trends demonstrate that the people in Lisbon are increasingly living in smaller households of one or two people.

This period has also witnessed a notable change in the age structure and sex composition of the population in different types of household. The proportion of the population living in *complex* households has decreased considerably. However, the decline was much more pronounced (22 percentage points between 1981 and 2011) for older women compared to all other age groups. It shows that in 1981 a large proportion of elderly women used to share dwellings with their children or other relatives, and in 2011 a large proportion of them live in *one-person* or *group quarter* households. In "*couple with children*" households, the proportion of both sexes in the adult age-group (15–64 years) has decreased significantly from 45.4% in 1981 to 39.8% in 2011. In contrast, in "*couples without children*" households, the proportion of adult population has increased from 12% in 1981 to 18% in 2011. It shows that many adults live together, but do not reproduce, which is in line with declining levels of fertility. The proportion of elderly in "*couples without children*" households also increased by about 10 percentage point during 1981–2011, reflecting the increases in life expectancy, especially for men who used to die much earlier than women. The proportion of women in "*group quarters*" has also increased significantly from 3.4% in 1981 to 4.3% in 2011. In 1981, the proportion of children living in *single-parent* households was 6%, which increased to 14.5% in 2011. The proportion of adult men and women in *single-parent* households also increased from 3.6% and 6.2% in 1981 to 8.5% and 13.1% in 2011, respectively. The proportion of women in *single-parent* households has increased remarkably, since in most cases of separation or divorce, child custody is awarded to mothers. The proportion of older people in these households has also increased slightly. The share of adult population of both sexes in *one-person* households has increased from 8% in 1981 to 13% in 2011. The most notable increase has been recorded by older women (65 years and over) from 28.1% in 1981 to 34.3% in 2011. The proportion of older men also increased from 11% in 1981 to 14% in 2011. These changes in household composition have a significant impact on the demand for housing in Lisbon.

Household formation depends primarily on the emancipation of youth and union formation (marital or consensual). Therefore, the proportion of young adults (20–34 years) living with their parents can be a good indicator showing the ease of nest-leaving in different contexts. In Lisbon, the proportion of young adults aged 30–34 years living with their parents has doubled from 11% in 1981 to 25.6% in 2001. After 2001, it decreased to 16.7% in 2011. Similarly, in the age group of 25–29 years, this share increased from 21.7% in 1981 to 44.8% in 2001. In 2011, it also declined to 33%. For the 20–24 age group, it has increased from 53% in 1981 to 62.5% in 2011. Women in each age group emancipate earlier than men. These trends are identical to those observed in other south European countries (Billari et al. 2002). The formation and dissolution of unions also play an important role in the formation of new households. Any new union formation is very likely to result in a new *couple without children* household. Similarly, dissolution of couples leads to a *single-parent* household or two *one-person* households. During the last few decades, people in Lisbon get married late and divorce more often. In 1981, the proportion of never married people in the total population was 40.3%, which increased to 45.3% in 2011. However, the proportion of separated or divorced people multiplied 4 times, from 2.6% in 1981 to 9.3% in 2011.

During the same period, the proportion of the widowed population has also increased by 1 percentage point from 7.7% in 1981 to 8.7% in 2011. Only the proportion of the married or in union population decreased from 49.4% in 1981 to 36.7% in 2011. All these changes have contributed to the increase in the number of *one-person* households and the disappearance of family households.

In summary, it can be concluded that despite a notable decrease in the size of Lisbon's population, the number of households has not decreased proportionally. However, the average size of households has decreased considerably. The proportion of *complex* households has decreased and that of *one-person* and *single-parent* households has increased notably. All of these changes have a significant impact on the demand for housing in Lisbon.

### 4.1.3. Housing Stock and Ownership

The term "housing stock" is commonly used to refer to the total number of dwellings[2] for domestic use in an area, region or country. The legacy of family financing, cultural values related to housing, public policies, privatization and the importance of housing in the welfare state played an important role in shaping the size and composition of housing stock in Portugal (Xerez and Fonseca 2016). According to the census data, in 1981, Lisbon had 239.8 thousand dwellings in its housing stock, which increased to 278.7 thousand in 1991, 293.2 thousand in 2001 and 323.9 thousand in 2011. It shows a steady increase in the number of dwellings in Lisbon. In 2011, of the total housing stock, 72.8% were exclusively residential apartments, 10.9% were secondary residences, 15.5% were vacant dwellings, 0.4% were non-residential buildings and the remaining 0.4% were hostels/hotels and group quarters. Over the past four decades, not only has the size of housing stock increased, its other characteristics such as age, the number of rooms per house and the quality of dwellings have also changed considerably. The current housing stock in Lisbon is the result of several construction cycles over the last century. In 2011, 67.3% of the population and 71.1% of all households lived in dwellings built before 1980. The average age of residential buildings in Lisbon has increased from 26 years in 1981 to 46 years in 2011. The rapid progress in technology and changes in demographic structure and lifestyles have changed the housing requirements of Lisbon's population that houses built four decades ago do not meet, which in turn creates various housing problems in the city.

Since 1981, the average size of dwellings in Lisbon's housing stock has also witnessed some notable changes. In the 1980s, the housing stock was much more diverse (in terms of the number of rooms) than in 2011. In 1981, the proportion of small-dwellings (less than 2 rooms) in the total housing stock was 20.9%, which decreased to 3.4% in 2011. During the same period, the proportion of medium-dwellings (3–4 rooms) also decreased from 54.8% in 1981 to 34.1% in 2011. However, the proportion of large dwellings (5+ rooms) increased considerably from 24.6% in 1981 to 62.2% in 2011. The shortage of small dwellings has a negative impact on the nest-leaving prospects of young adults, who cannot afford large dwellings. Furthermore, as the age of residential buildings increases, their potential to maintain a comfortable and healthy life decreases, especially for older people. Since more than two-thirds of the residential buildings in the municipality of Lisbon were constructed before 1980, a significant part of them lacks basic services such as central heating system, air-conditioning and most importantly, elevators, in the case of multi-floor buildings. According to the 2011 census, 16.1% of all dwellings in Lisbon were without air-conditioning, 15.1% were without heating system and 44.3% were without elevator. The lack of these basic amenities has several negative consequences for the inhabitants of these substandard dwellings. It is a major concern for an aging population.

Regarding homeownership, like other southern European countries, e.g., Italy and Spain, the high rate of homeownership has been a prominent feature of the Portuguese housing market (Azevedo 2016). In Portugal, housing has been financed: "through public

---

[2] According to Portuguese census data, "a dwelling is an enclosed and independent place that is built, rebuilt, expanded, transformed or used exclusively for living purposes during the reference period" (IPUMS, Portuguese Census 2011).

funds in the case of social housing, through family financing and through bank loans"
(Xerez and Fonseca 2016). Especially, the role of family in financing of housing through
self-promotion is very crucial (Fahey and Norris 2011; Minas et al. 2013). According to the
2011 census, in Portugal 74% of the dwellings used as main residence were owner-occupied,
18.2% were private rental or sub-rented, and 1.7% were under the social housing schemes.
In fact, in Portugal, the social housing stock is very small and its access is means-tested,
which means that only families with high levels of socioeconomic vulnerability have
access to it (Alves and Andersen 2019). The remaining 6.8% of the dwellings referred to
situations other than those described above (e.g., accommodation provided for free). The
circumstances in Lisbon differ from the national context, with the private and social rental
markets being more relevant than at the national level. In 2011, 51.8% of the dwellings used
as a main residence were owner-occupied, 36% were rented or sub-rented in the private
market, 6.2% referred to social housing, and 6% to other situations.

Contrary to national trends, the proportion of owner-occupied dwellings in Lisbon
was merely 17.6% in 1981. In the 1980s, the political response to the housing shortage
was to encourage homeownership, which influenced the economic and institutional envi-
ronment of the housing market and its regulations to facilitate access to homeownership
(Xerez and Fonseca 2016). Consequently, homeownership became the main investment
for most households in Lisbon (Neves 2000). In the following decades, the proportion of
homeowners increased considerably to 33.5% in 1991, 47.1% in 2001 and 51.8% in 2011.
Currently, more than half of the households have owned dwellings, which reduces the
flexibility of the housing market to meet the needs of the rapidly changing population
and households. More recently, the real estate market in Portugal has started to witness
signs of housing overvaluation due to the recovery of the Portuguese economy from the
economic crisis, the growth of short-term rental market and the interest of foreign investors
in the Portuguese real estate market (Banco de Portugal 2018). The strong preference
for becoming a homeowner and the speculative investments in the housing market has
notably increased the sale price of houses in Lisbon. According to the INE's Housing Price
Index, the average value of dwellings in Lisbon metropolitan area has increased from 150.4
thousand in 2009 to 209.4 thousand in 2020 (INE 2021a). Similarly, the median value of
per square meter of dwellings in the municipality of Lisbon has increased from EUR 1841
in the second quarter of 2016 to EUR 3227 per square meter in the second quarter of 2020
(INE 2021b).

As per the long-term rental market, the rural exodus of the 1960s and the immigration
caused by the decolonization process in the 1970s increased the demand for long-term
rentals in Lisbon, generating marked social tensions between natives and immigrants
(Pinto 2008; Agarez 2018). The rent freezing legislation of the mid-1960s had made the
rental market very unattractive to homeowners in Portugal (Perista and Baptista 2007). It
affected the supply of long-term rentals in the housing market and increased the rent of
long-term rentals in the city. The 2007–2008 global economic crisis, which led Portugal
to request for financial assistance from the European Commission, the European Central
Bank and the International Monetary Fund, marked the beginning of austerity policies
that liberalized the long-term rental market, relaxed the land-use regulations and created a
tax regime favorable to real estate speculations, which aggravated the problem of access
to housing in Lisbon and other major cities in Portugal (Allegra and Tulumello 2019). A
detailed examination of the long-term rental market in Lisbon shows that the monthly rent
in 65% of the total rent contracts signed before 1975 was less than EUR 100 and in only
2.4% of the contracts the monthly rent exceeded EUR 650. In the 2006–2011 period, the
proportion of rental contracts of less than EUR 100 decreased to 4.12% and that of above
EUR 650 increased to 27.8% (Figure 2). This shows a considerable increase in monthly
house rents in Lisbon. According to recent estimates by INE, the median price of new
rental contracts has increased by 21.2% in Portugal and 24.3% in Lisbon between 2017 and
2019 (INE 2021a).

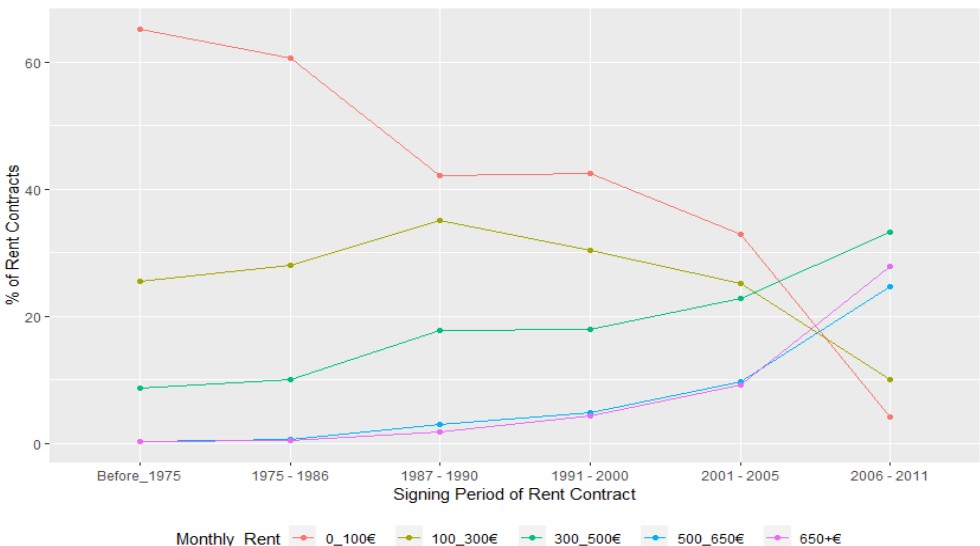

**Figure 2.** The growth of average monthly rent in Lisbon. Source: Own elaboration, with data from Census records 2011.

### 4.2. Population and Housing Mis(matches) in Lisbon

In the past four decades, the rapid transformation of demographics (population structure and household composition) and changes in housing (stock and tenancy regime) have created some mismatches in Lisbon's housing market. Despite a considerable decrease in the size of its population and the number of households, rampant construction from the beginning of the democratic period (after the 1974 revolution) to the 2007–2008 global economic crisis, has created a surplus of dwellings in Lisbon. It resulted in a mismatch between the actual demand and supply of housing units, where in 2011 the number of dwellings was 32.5% higher than the number of households in Lisbon (Figure 3). This excessive housing stock consists of vacant dwellings, which includes inherited dwellings for resale, apartments held by real estate investors and banks for resale, and second homes, which are occasionally used for vacations (Xerez and Fonseca 2016). This stock of vacant houses, on the one hand, has increased the built-up area in Lisbon, which in turn has several negative consequences for the environment and the quality of life in the city, on the other hand, incurring a huge annual public and private expenditure on the maintenance of vacant buildings and the provision of basic services such as electricity, water supply and sewerage in the neighborhoods where they are located. The average age of Lisbon's population has been increasing and the fertility is decreasing for the last four decades. Following the arguments raised by Lindh and Malmberg (1999, 2008), with less and less people entering in the age of nest leaving, the demand for new houses in Lisbon is bound to decrease in the coming years. In that case, these excessive vacant apartments will continue to be a great burden for the local government.

Over the last four decades, the average size of households in Lisbon has decreased considerably from 2.8 people per household in 1981 to 2.2 in 2011. At the same time, the share of large dwellings in the housing stock has increased from 24.6% in 1981 to 62.2% in 2011 (Figure 4). It shows that the construction waves and real estate planners had completely ignored the household dynamics of Lisbon. Consequently, the share of small dwellings in the housing stock has been reduced from 21% in 1981 to 3% in 2011. Since the change in household size has a direct impact on housing needs (Mulder 2006), it has created a mismatch between the demand and supply of dwellings of different sizes in Lisbon. Currently, the shortage of small and affordable houses makes it difficult for young adults to rent a suitable dwelling for themselves and form new *one-person* or *couple without children* households. Consequently, in 2011, more than 11 percent of all young adults (aged 30–34 years) in Lisbon lived in their parental homes. The growing discontent among young

people is culminating in the form of new struggles and social movements for the right to housing in Lisbon (Tulumello 2019).

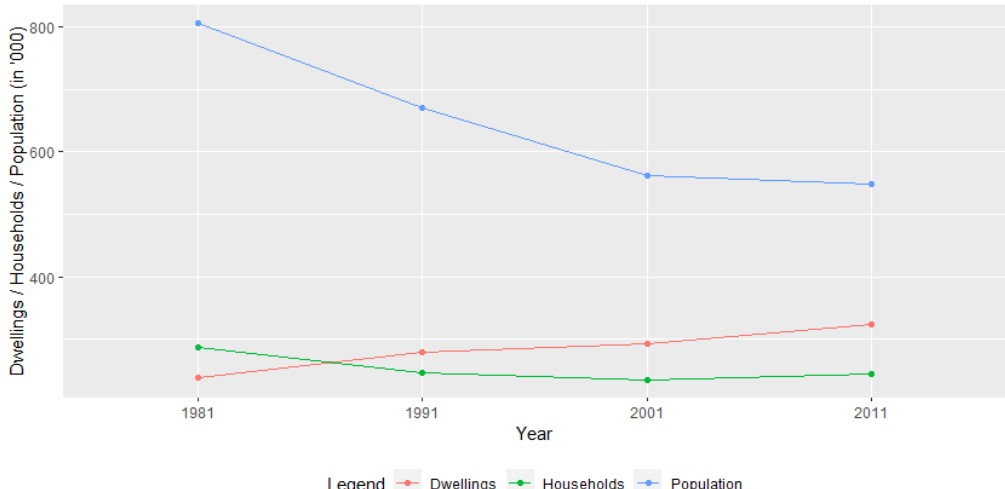

**Figure 3.** The evolution of family dwellings, household and population in Lisbon, 1981–2011. Source: Own elaboration, with data from Census records 1981–2011.

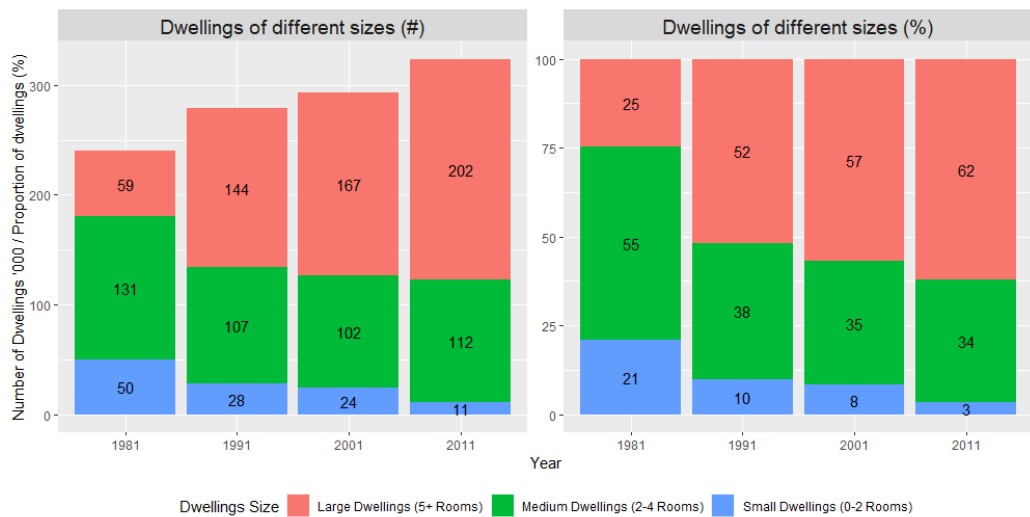

**Figure 4.** Family dwellings size in Lisbon, 1981–2011. Source: Own elaboration, with data from Census records 1981–2011.

This imbalance between supply and demand for housing also leads to simultaneous over- and under-consumption of dwellings in Lisbon. Due to patterns of low residential mobility, a characteristic of the Portuguese housing system (Azevedo 2020), some elderly people live in dwellings larger than necessary for the size of their household, while, at the same time, disadvantaged groups such as migrant families and young adults live in shared and/or small dwellings. As a result of the female advantage in life expectancy, the majority of *one-person* households in large dwellings are owned by older women (Figure 5). This drive to "aging in place" (Frey 2007) among Lisbon's older women, who mostly own large dwellings, reduces household mobility and makes the housing market less responsive to changes in demographic and household structure. This mismatch affects the supply of large dwellings in the housing market and adds constraints for young couples who want to form families and start having children.

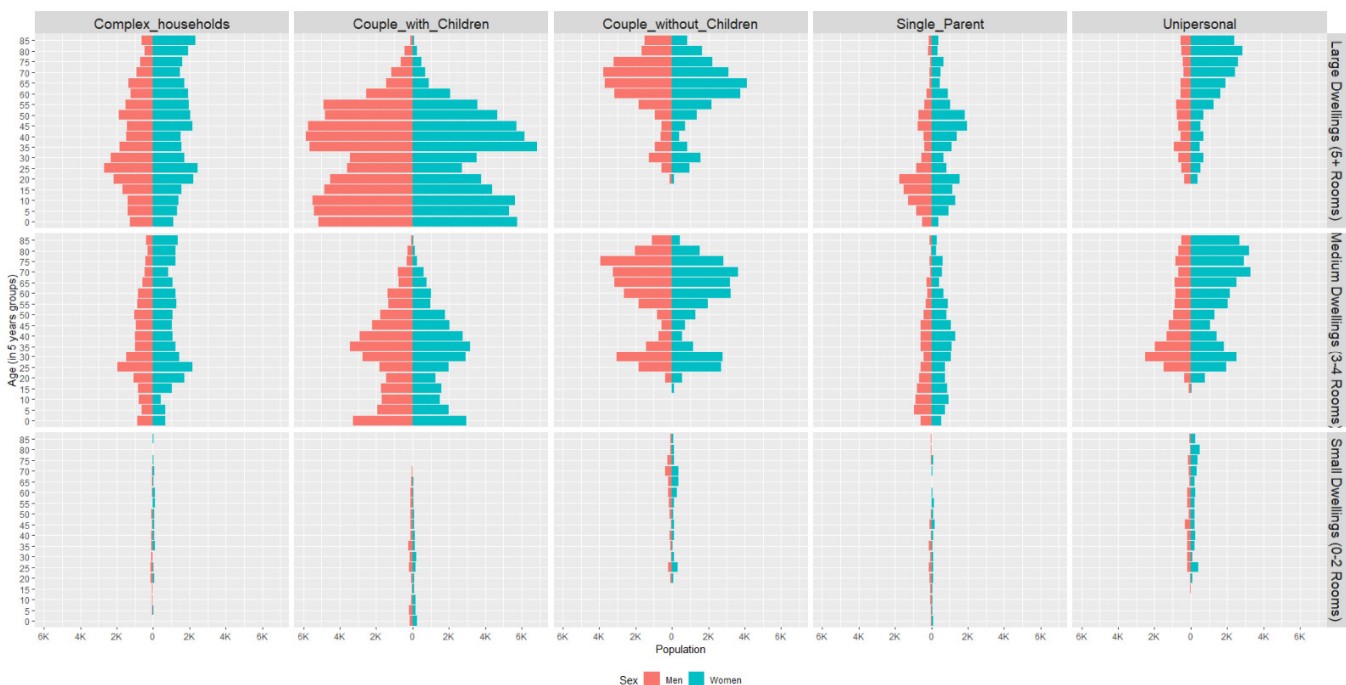

**Figure 5.** Relationship between the size of dwellings and household type, 2011. Source: Own elaboration, with data from Census records 2011.

The quality of housing in Lisbon is also a major concern. According to 2011 Census data, 11.4% of Lisbon's dwellings needed at least medium repairs. This proportion is even higher among tenants in the private sector (16.8%) and the public housing sector (21.8%). The elderly and disabled are more vulnerable to the lack of essential services such as heating, air-conditioning, and elevators. In Lisbon, there is a mismatch between the quality of dwellings and the requirements of inhabitants, leading to many people living in substandard housing. In 2011, 86.8% of the total population over 65 in Lisbon had dwellings without air-conditioning, 15.7% without heating system, and 50.7% without elevators (Figure 6). All of these deficiencies compromise the health and wellbeing of the elderly (OECD 2003). As the number of people over 65 increases rapidly with a constant increase in life expectancy at birth, this mismatch will create more problems for the elderly population in Lisbon in the coming decades. In fact, a large number of disability problems in the elderly, which incurs large health and social costs, are also related to accidents associated with falls. In addition to the elderly, many of these substandard houses are occupied by young adults in the 20–35 age group, and most of them are immigrants from different non-EU countries.

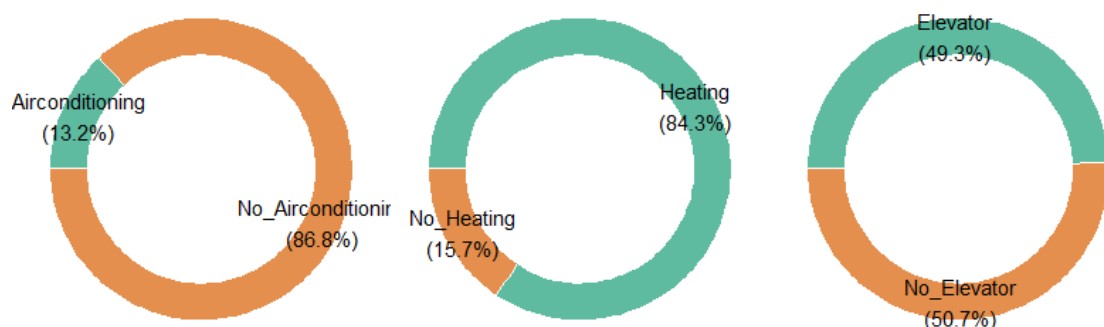

**Figure 6.** Basic services in the dwellings occupied by 65 years and over people in Lisbon, 2011. Source: Own elaboration, with data from Census records 2011.

In Lisbon, the size of the housing stock is much larger than the number of house-holds, creating an excess supply of housing. Normally, the supply is inversely related to price—when the supply increases, the price goes down—but in Lisbon, despite the large supply of dwellings, the monthly rents and the purchase prices increase continually. Contrary to the findings of Pitkin and Myers (2008), despite the rapid aging of the population since 1981, the price of housing (mortgages and monthly rents) in Lisbon has increased considerably (for more detail, see Xerez et al. 2019). This growth trend intensified in the aftermath of the 2007–2008 economic crisis, making it difficult for middle-class people to buy or rent homes in Lisbon (Mendes 2017). The constitutional right to adequate housing has become a luxury for many people in Lisbon. Between the second quarter of 2017 and 2019, the median value of rents per square meter of new rental contracts for family accommodation in Lisbon grew by 24.3% (INE 2021a). Similarly, between the last quarter of 2017 and 2019, the median value of sales per square meter of housing units in existing apartments grew by 32.4% (INE 2021b), while the average real income of households is stagnant. This overvaluation of family dwellings is the result of the recovery of the Portuguese economy, the increase in tourism and the consequent increase in tourist accommodation and the interest of non-residents in the Portuguese real estate market (Banco de Portugal 2018). It has created a huge mismatch between the purchasing power of the local population and the price of housing in Lisbon.

Similarly, a high preference for homeownership, a loosely regulated rental market, and difficulty in obtaining home loans have created new problems for adequate and affordable housing for Lisbon's population, which is becoming increasingly diverse due to the continued immigration of people from different origins. The growing proportion of immigrant population, lack of long-term rentals and difficulties in obtaining mortgage loans have led to a situation in which rental prices are currently higher than the actual monthly mortgages (Azevedo 2020).

## 5. Conclusions

This article highlights several mismatches between the demographic structure and the housing stock in Lisbon. These mismatches are caused by an imbalance between the rapid demographic transition and changes in the housing stock in the last four decades that do not correspond to the needs of the resident population. Currently, the number of dwellings in Lisbon exceeds households, so there is no mathematical need to build more residential buildings. In addition, various policy instruments at the local and national level that, theoretically, should be contributing to ensure access to housing for all (e.g., "Rehabilitate to Rent-Affordable Housing" and "Affordable Rental Program") are underway. However, extensive work is still required to renovate the current housing stock.

In Lisbon, the main problem of the housing stock is its poor quality. The suitability of the housing stock is of vital importance for people's health and well-being. As the number of people aged 65 and over in Lisbon's population increases, the suitability of the housing stock is becoming a major health and safety concern. A large number of disability problems among older people are due to accidents that occur in their residential buildings (e.g., falls). Therefore, a significant part of the housing stock, which is in a substandard condition for a healthy life, must be rebuilt or rehabilitated with public–private collaboration and placed in social housing for vulnerable groups such as single-parents, disabled and elderly people in the city. Rehabilitation of substandard dwellings owned and occupied by older people must be carried out with the help of public interventions.

For the proper functioning of the housing market, the housing stock in Lisbon must be diversified to meet the needs of different age and socioeconomic groups. Lisbon currently has an oversupply of large houses, which are costly to maintain and better suited for large families; however, there is an acute shortage of small budget apartments, which are affordable and more adequate for *one-person* households or young adults who want to emancipate. Adequate and affordable houses of different sizes and with different tenancy

regimes should be provided to maintain a balance in the demand and supply of housing in Lisbon's housing market.

Residential buildings that are too old for rehabilitation must be rebuilt in such a way that they can be adapted when necessary. All new construction projects must be carried out to stimulate the development of lifetime housing that can adapt to the needs of the inhabitants and reduce their dependence on others. To understand their needs and desires, the elderly population must participate in the specification, design, planning and execution of new housing projects. Public investments should be made to provide desirable accommodation in suitable locations for retirees who wish to live in retirement housing in later life. Private sector retirement housing projects should be encouraged and supported to create housing diversity and help meet the future housing needs of older homeowners.

A maximum rent limit (similar to the "Rent Price Cap law" in Berlin) should be set for apartments of different sizes in different areas to discourage speculative increase in rents. At the same time, hoarding of residential apartments by banks and other financial institutions could be restricted by imposing a high property tax on vacant apartments. The property tax could be increased on the second homes in the city. The spread of short-term rentals in the central neighborhoods of the city should be controlled by strict laws and homeowners should be encouraged to rent their apartments on long-term leases.

The population is changing, now more with migration than with natural growth. Immigrants, especially those who migrate in search of work and better living conditions, tend to be young. They can rejuvenate and diversify Lisbon's population, which can eventually increase the demand for housing in the city. All these mismatches can aggravate the housing problem of these newcomers to Lisbon, since most of them are single and lack the resources to buy or rent large apartments in the city. In the future, it will be interesting to speculate on how the population and household structure in Lisbon will evolve over time and what changes need to be made to the existing housing stock and tenancy regime to meet the requirements of future residents of Lisbon.

**Author Contributions:** Conceptualization, N.S.G. and A.B.A.; methodology, N.S.G. and A.B.A.; software, N.S.G. and A.B.A.; validation, N.S.G. and A.B.A.; formal analysis, N.S.G. and A.B.A.; investigation, N.S.G. and A.B.A.; resources, N.S.G. and A.B.A.; data curation, N.S.G. and A.B.A.; writing—original draft preparation, N.S.G. and A.B.A.; writing—review and editing N.S.G. and A.B.A.; visualization, N.S.G. and A.B.A.; supervision, N.S.G. and A.B.A.; project administration, N.S.G. and A.B.A.; funding acquisition N.S.G. and A.B.A. All authors have read and agreed to the published version of the manuscript.

**Funding:** This research was funded by Fundação para a Ciência e Tecnologia, Nachatter Singh Garha in project HOPES: Housing Perspectives and Struggles. Futures of housing movements, policies and dynamics in Lisbon and beyond, grant number: PTDC/GESURB/28826/2017; Alda Botelho Azevedo in project SUSTAINLIS: Sustainable urban requalification and vulnerable populations in the historical centre of Lisbon, Grant number: PTDC/GESURB/28853/2017.

**Institutional Review Board Statement:** Not applicable.

**Informed Consent Statement:** Not applicable.

**Data Availability Statement:** IPUMS International Database, Portuguese Census, 2011.

**Conflicts of Interest:** The authors declare no conflict of interest.

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
