# Peer review of "Population and Housing (Mis)match in Lisbon, 1981–2018. A Challenge for an Aging Society"

_socsci, doi:10.3390/socsci10030102_

Round 1

Reviewer 1 Report

Dear Author(s),

The aim of paper entitled “ Population and Housing (mis)match in Lisbon, 1981-2018. A challenge for an aging society” was to reserch the difference between population (or household) and housing dynamics. It is an interesting article showing the problem between the demand and supply of adequate and affordable dwellings for the people from different age groups and socio-economic classes in Lisbon. After reviewing this article, I have comments and suggestions as follows:

SOURCES AND METHODS: I sugest to move the figure 1 [on line 100], to the third section describes the data sources and methods. At the same time, the presented figure 1 should be discussed, in the context of the structure of the paper (section of work). After all, I suggest clearly presenting the indicators adopted for yours research, Maybe putting them in the table would be a good solution ?.

RESULTS : In my opinion the interpretation of the results is too long and not transparent. Maybe it is worth presenting some statistical data in a table and commenting on the most important issues.

OTHERS: There is a reference in the text to figure 7 [line 359] and figure 8 [371] which are not there.

Author Response

Response to Reviewer 1

We would like to thank the reviewer for his/her thoughtful review of the manuscript. He/she has raised important issues and his/her inputs are very helpful for improving the manuscript. We have revised some parts of our manuscript. Below, we describe the ways in which we dealt with the reviewers’ comments.

The aim of paper entitled “ Population and Housing (mis)match in Lisbon, 1981-2018. A challenge for an aging society” was to research the difference between population (or household) and housing dynamics. It is an interesting article showing the problem between the demand and supply of adequate and affordable dwellings for the people from different age groups and socio-economic classes in Lisbon. After reviewing this article, I have comments and suggestions as follows:

SOURCES AND METHODS: I sugest to move the figure 1 [on line 100], to the third section describes the data sources and methods. At the same time, the presented figure 1 should be discussed, in the context of the structure of the paper (section of work). After all, I suggest clearly presenting the indicators adopted for yours research, Maybe putting them in the table would be a good solution.

  • We are not in favour of moving the figure to data source section, as this figure provides the theoretical framework for the study.
  • Some information is added to the existing table about the demographic transformation of the city.

RESULTS : In my opinion the interpretation of the results is too long and not transparent. Maybe it is worth presenting some statistical data in a table and commenting on the most important issues.

  • The result section has been revised.

OTHERS: There is a reference in the text to figure 7 [line 359] and figure 8 [371] which are not there.

Corrected. These figures were removed from the main text due to the limited space.

Reviewer 2 Report

Dear Authors,

The article is interesting. It discusses important and valid problems regarding the ageing of society, and in particular the dependencies between: 1) changes in the demographic structure of the population and the housing market and 2) the existence of different (mis)matches between the housing needs of population (or households) and the available housing stock in Lisbon. The study design is appropriate, the literature review is very good. However, there are some details that should be improved prior to publication:

  • Major revision
  1. Please, indicate the added value of the article in the abstract.
  2. It lacks research questions or hypotheses in the Introduction. Also, please, indicate the added value of the article.
  3. Please, justify the choice of Lisbon as a geographic research area.
  4. The discussion is weak – Authors should compare the results obtained with the arguments presented in the theoretical part and expand them further. An international comparison would also be appropriate. Are there similar trends in other countries (European capitals) or is the capital of Portugal specific in the terms of the problem studied? What could an international reader learn from it? Practical implications should be expanded.
  5. Taking into account the need to expand Discussion part, it seems reasonable to name point 4 "Results and Discussion" and point 5 “Conclusion”.

  • Minor revision
  1. The Authors refer to figure 7 (line 359) in the text, but there is no such graph in the article.
  2. The Authors refer to figure 8 (line 371) in the text, but there is no such graph in the article.
  3. Point 4.1 appears twice in the article. (lines: 213, 421).
  4. There is no description of the colours in the figure 4.

Author Response

Response to Reviewer 2

We would like to thank the reviewer for his/her thoughtful review of the manuscript. He/she has raised important issues and his/her inputs are very helpful for improving the manuscript. We have revised some parts of our manuscript. Below, we describe the ways in which we dealt with the reviewers’ comments.

Major revision

Please, indicate the added value of the article in the abstract.

Done.

It lacks research questions or hypotheses in the Introduction. Also, please, indicate the added value of the article.

Done.

Please, justify the choice of Lisbon as a geographic research area.

Done.

The discussion is weak – Authors should compare the results obtained with the arguments presented in the theoretical part and expand them further. An international comparison would also be appropriate. Are there similar trends in other countries (European capitals) or is the capital of Portugal specific in the terms of the problem studied? What could an international reader learn from it? Practical implications should be expanded.

  • We have made changes in the manuscript to link the theoretical part with the discussion.
  • Comparison between Lisbon and rest of Portugal, and other cities has been done.

Taking into account the need to expand Discussion part, it seems reasonable to name point 4 "Results and Discussion" and point 5 “Conclusion”.

 Done.

Minor revision

The Authors refer to figure 7 (line 359) in the text, but there is no such graph in the article.

removed

The Authors refer to figure 8 (line 371) in the text, but there is no such graph in the article.

removed

Point 4.1 appears twice in the article. (lines: 213, 421).

corrected

There is no description of the colours in the figure 4

Description added

Reviewer 3 Report

This paper investigates the existence of a potential mismatch in the Lisbon housing market. The drivers that are explored to understand the outcomes of the housing market are the demographics elements. This paper is interesting, however, the authors would need to do a major revision to bring the current manuscript into publishable material.

Comments:

    1. Keywords: I would not use ‘aging’, ‘population’ and ‘households’ as keywords. They are too generic. I propose to used: Housing stock, Demographics, Lisbon. Apart from that I would like the authors to add 2 new keywords that are more meaningful and reflect in a better way the content of the paper.
    2. Line 36-68: I found these lines more suitable for the literature review part than for the introduction. I suggest to condense them.
    3. The introduction should be rewritten. I found it too long and difficult to see what is the value added of this paper. Why are the authors focusing on Lisbon? Why is this so important? How are they going to do their analysis? I found that this questions are not properly responded in the current version.
    4. Line 69: Use ‘testable hypothesis’ instead of ‘main argument’.
    5. Please, refer to a contribution that presents a conceptual framework to explain the complex relationship between housing and demographics. An example could be: Arestis, P. and González, A.R. (2015), ‘Importance of Demographics for Housing in the OECD Economies’, Bulletin of Economic Research.
    6. Line 72: The less flexible housing stock […]. Please, rewrite the sentence as follows: The lack of flexibility in the housing stock (new construction, rehabilitation) and a rigid regulatory rent regime has failed to keep pace with a highly dynamic population and the current household structures.
    7. Line 77: Please, authors add a sentence to briefly introduce the methods that are used to ‘identify the existence of different (mis)matches’ in the Lisbon market
    8. Lines 78-84: Some editing of this paragraph is needed. I would suggest the following: The paper is structured as follows. Section 2 presents a detailed review of existing studies exploring the relationship between demographics and housing. Section 3 describes the data sources and methods used in this study. Section 4 analyses population and household structure, size and quality of Lisbon’s housing stock. It also highlights the mismatches between population and housing dynamics in Lisbon. Finally, Section 5 presents some concluding remarks and a discussion regarding the main aspects of these mismatches and suggests some measures to deal with them.
    9. Line 119-121: The statement: ‘It can lead to a situation in which the supply of housing units exceeds their demand, resulting into underutilisation of housing stock and a fall in home equity (Pitkin and Myers, 2008)’ is an interesting one. However, the argument is not completely explored. I am missing some text to explain the price effects related to this.
    10. Line 140: It is the first time that the authors mentioned a potential relationship between unemployment and housing demand. This topic is very important to understand the dynamics of the housing market and it is not always well acknowledge in the literature. There is also some controversy regarding the direction of this relationship. So, I recommend the authors to add a footnote with some background on the topic and refer to: Arestis, P. and González, A.R. (2019). “Economic Precariousness: A New Channel in the Housing Market”, International Journal of Finance and Economics. https://doi.org/10.17863/CAM.35973. This paper could also provide them with other relevant references to add.
    11. Line 192: The authors are listing a variety of factors that drive the housing market. One or two references more should be added to Herbert et al. (2005).
    12. ‘However, this paper only focuses on the two-sided relationship between demographic factors and housing characteristics that affect the functioning of the housing market in Lisbon.’ Please, authors provide 2 or 3 sentences to explain why it is so important to focus on the Lisbon market. Besides, this text (and the new one) should be placed in the introduction.
    13. References along the text should be cited in a consistent manner.
    14. Data sources and methods: Please, add a footnote with a link to the source of the data.
    15. Line 449-451: ‘Currently, the shortage of small houses makes it difficult for young adults to rent a suitable dwelling for them and to form new one-person or couple without children households.’ How large is the share of this group of population? Is it expected to increase?
    16. Line 469: The quality of housing in Lisbon is also a major concern. Please, refer to a report/paper that deals with this.
    17. Line 474: All these deficiencies compromise the health and wellbeing of the elderlies. Please refer to some literature that elaborates on this topic.
    18. The link to housing policy in Portugal/Lisbon is totally missing. I recommend the authors to add a subsection at the beginning of Section 4. In this section I am also missing some content comparing the Lisbon reality with the rest of the country. For a non-Portuguese reader is difficult to place this in the broader country context.
    19. There are several places in which the authors refer to ‘affordability’. However, no data on prices or affordability ratios are provided in Section 4. This should be corrected.
    20. In section 5, the authors should highlight which are the policy implications of their research. Which lessons could be extrapolated to other cities.
    21. Line 528-532: ‘In the future, it will be interesting to speculate on how the population and household structure in Lisbon will evolve over time and what changes need to be made to the existing housing stock to meet the requirements 530 of future residents of Lisbon.’ Indeed, it would be interesting, so I would suggest to add a final paragraph exploring this topic.
    22. Further editing of the text would be beneficial.

Author Response

Response to Reviewer

We would like to thank the reviewer for his/her thoughtful review of the manuscript. He/she has raised important issues and his/her inputs are very helpful for improving the manuscript. We have revised some parts of our manuscript. Below, we describe the ways in which we dealt with the reviewers’ comments.

This paper investigates the existence of a potential mismatch in the Lisbon housing market. The drivers that are explored to understand the outcomes of the housing market are the demographics elements. This paper is interesting, however, the authors would need to do a major revision to bring the current manuscript into publishable material.

Comments:

Keywords: I would not use ‘aging’, ‘population’ and ‘households’ as keywords. They are too generic. I propose to used: Housing stock, Demographics, Lisbon. Apart from that I would like the authors to add 2 new keywords that are more meaningful and reflect in a better way the content of the paper.

  • We have added two new keywords to the manuscript replacing previous ones.

Line 36-68: I found these lines more suitable for the literature review part than for the introduction. I suggest to condense them.

  • Done

The introduction should be rewritten. I found it too long and difficult to see what is the value added of this paper. Why are the authors focusing on Lisbon? Why is this so important? How are they going to do their analysis? I found that this questions are not properly responded in the current version.

  • All these issues are responded in detail in the introduction part of the manuscript

Line 69: Use ‘testable hypothesis’ instead of ‘main argument’.

Please, refer to a contribution that presents a conceptual framework to explain the complex relationship between housing and demographics. An example could be: Arestis, P. and González, A.R. (2015), ‘Importance of Demographics for Housing in the OECD Economies’, Bulletin of Economic Research.

  • Done

Line 72: The less flexible housing stock […]. Please, rewrite the sentence as follows: The lack of flexibility in the housing stock (new construction, rehabilitation) and a rigid regulatory rent regime has failed to keep pace with a highly dynamic population and the current household structures.

  • Done

Line 77: Please, authors add a sentence to briefly introduce the methods that are used to ‘identify the existence of different (mis)matches’ in the Lisbon market

  •  

Lines 78-84: Some editing of this paragraph is needed. I would suggest the following: The paper is structured as follows. Section 2 presents a detailed review of existing studies exploring the relationship between demographics and housing. Section 3 describes the data sources and methods used in this study. Section 4 analyses population and household structure, size and quality of Lisbon’s housing stock. It also highlights the mismatches between population and housing dynamics in Lisbon. Finally, Section 5 presents some concluding remarks and a discussion regarding the main aspects of these mismatches and suggests some measures to deal with them.

  • Done

Line 119-121: The statement: ‘It can lead to a situation in which the supply of housing units exceeds their demand, resulting into underutilisation of housing stock and a fall in home equity (Pitkin and Myers, 2008)’ is an interesting one. However, the argument is not completely explored. I am missing some text to explain the price effects related to this.

  • More information about housing prices is added to the manuscript.

Line 140: It is the first time that the authors mentioned a potential relationship between unemployment and housing demand. This topic is very important to understand the dynamics of the housing market and it is not always well acknowledge in the literature. There is also some controversy regarding the direction of this relationship. So, I recommend the authors to add a footnote with some background on the topic and refer to: Arestis, P. and González, A.R. (2019). “Economic Precariousness: A New Channel in the Housing Market”, International Journal of Finance and Economics. https://doi.org/10.17863/CAM.35973. This paper could also provide them with other relevant references to add.

  • Done

Line 192: The authors are listing a variety of factors that drive the housing market. One or two references more should be added to Herbert et al. (2005).

‘However, this paper only focuses on the two-sided relationship between demographic factors and housing characteristics that affect the functioning of the housing market in Lisbon.’ Please, authors provide 2 or 3 sentences to explain why it is so important to focus on the Lisbon market. Besides, this text (and the new one) should be placed in the introduction.

  • Done

References along the text should be cited in a consistent manner.

Data sources and methods: Please, add a footnote with a link to the source of the data.

  • Done

Line 449-451: ‘Currently, the shortage of small houses makes it difficult for young adults to rent a suitable dwelling for them and to form new one-person or couple without children households.’ How large is the share of this group of population? Is it expected to increase?

  • Done

Line 469: The quality of housing in Lisbon is also a major concern. Please, refer to a report/paper that deals with this.

  • Done

Line 474: All these deficiencies compromise the health and wellbeing of the elderlies. Please refer to some literature that elaborates on this topic.

  • Done

The link to housing policy in Portugal/Lisbon is totally missing. I recommend the authors to add a subsection at the beginning of Section 4. In this section I am also missing some content comparing the Lisbon reality with the rest of the country. For a non-Portuguese reader is difficult to place this in the broader country context.

  • Done

There are several places in which the authors refer to ‘affordability’. However, no data on prices or affordability ratios are provided in Section 4. This should be corrected.

  • Done

In section 5, the authors should highlight which are the policy implications of their research. Which lessons could be extrapolated to other cities.

  • Done

Line 528-532: ‘In the future, it will be interesting to speculate on how the population and household structure in Lisbon will evolve over time and what changes need to be made to the existing housing stock to meet the requirements 530 of future residents of Lisbon.’ Indeed, it would be interesting, so I would suggest to add a final paragraph exploring this topic.

  • We are writing a complete paper on that soon it will be ready

Further editing of the text would be beneficial.

  • Done

Round 2

Reviewer 2 Report

Dear authors,

thank you for incorporating my comments. The paper has improved from the previous version. Almost all my comments have been addressed.

Author Response

Response to Reviewer

We would like to thank the reviewer for his/her thoughtful review of the manuscript. He/she has raised important issues and his/her inputs are very helpful for improving the manuscript. We have revised some parts of our manuscript. Below, we describe the ways in which we dealt with the reviewers’ comments.

Comment 1

Reviewer: There are several places in which the authors refer to ‘affordability’. However, no data on prices or affordability ratios are provided in Section 4. This should be corrected.

Authors: Done

Reviewer (New comment): I think that understanding price dynamics is important. I am still missing a chart with house and rental prices. I would suggest the authors to introduce a subsection to present there the information on price evolution, providing also the chart that I have just indicated.

Author: We have added some new information about the evolution of rent and sale prices of houses in Lisbon.

Comment 2

Reviewer: Line 528-532: ‘In the future, it will be interesting to speculate on how the population and household structure in Lisbon will evolve over time and what changes need to be made to the existing housing stock to meet the requirements 530 of future residents of Lisbon.’ Indeed, it would be interesting, so I would suggest to add a final paragraph exploring this topic.

Authors: We are writing a complete paper on that soon it will be ready

Reviewer (New Comment): This is good news. However, I would suggest to close the article in a different way. The final paragraph should be replaced. Maybe the authors can write something along the following lines: ‘The population is changing, now more with migration than with natural growth. Migrants who search for a job and better living conditions tend to be young. They can rejuvenate Lisbon population, which can eventually increase the demand for housing in the city. [Here I would suggest the authors to add 2 sentences to give a hint on the implications of this change in demand for the mismatches that authors were reporting. A final sentence to indicate that the topic will be further explored in a future contribution should be added].

Author: We have added a sentences to the final paragraph.

Comment 3

Reviewer (New Comment): Please, do further editing of the text. There are some typos.

Author: Further editing has been done and typos are corrected.

Reviewer 3 Report

I am pleased to see that the article is now progressing in the good direction. However, I still have some comments.

Comment 1

  • Reviewer: There are several places in which the authors refer to ‘affordability’. However, no data on prices or affordability ratios are provided in Section 4. This should be corrected.
  • Authors: Done
  • Reviewer (New comment): I think that understanding price dynamics is important. I am still missing a chart with house and rental prices. I would suggest the authors to introduce a subsection to present there the information on price evolution, providing also the chart that I have just indicated.

Comment 2

  • Reviewer: Line 528-532: ‘In the future, it will be interesting to speculate on how the population and household structure in Lisbon will evolve over time and what changes need to be made to the existing housing stock to meet the requirements 530 of future residents of Lisbon.’ Indeed, it would be interesting, so I would suggest to add a final paragraph exploring this topic.
  • Authors: We are writing a complete paper on that soon it will be ready
  • Reviewer (New Comment): This is good news. However, I would suggest to close the article in a different way. The final paragraph should be replaced. Maybe the authors can write something along the following lines: ‘The population is changing, now more with migration than with natural growth. Migrants who search for a job and better living conditions tend to be young. They can rejuvenate Lisbon population, which can eventually increase the demand for housing in the city. [Here I would suggest the authors to add 2 sentences to give a hint on the implications of this change in demand for the mismatches that authors were reporting. A final sentence to indicate that the topic will be further explored in a future contribution should be added].

Comment 3

  • Reviewer (New Comment): Please, do further editing of the text. There are some typos.

Author Response

(The authors gave the same response as above.)
